# DIFFUSION GENERATIVE MODELS ON SO(3)

## ABSTRACT

Diffusion-based generative models represent the current state-of-the-art for image generation. However, standard diffusion models are based on Euclidean geometry and do not translate directly to manifold-valued data. In this work, we develop extensions of both score-based generative models (SGMs) and Denoising Diffusion Probabilistic Models (DDPMs) to the Lie group of 3D rotations, SO(3). SO(3) is of particular interest in many disciplines such as robotics, biochemistry and astronomy/planetary science. Contrary to more general Riemannian manifolds, SO(3) admits a tractable solution to heat diffusion, and allows us to implement efficient training of diffusion models. We apply both SO(3) DDPMs and SGMs to synthetic densities on SO(3) and demonstrate state-of-the-art results.

## 1 INTRODUCTION

Deep generative models (DGM) are trained to learn the underlying data distribution and then generate new samples that match the empirical data. There are several classes of deep generative models, including Generative Adversarial Networks (Goodfellow et al., 2014), Variational Auto Encoders (Kingma & Welling, 2013) and Normalizing Flows (Rezende & Mohamed, 2015). Recently, a new class of DGMs based on Diffusion, such as Denoising Diffusion Probabilistic Models (DDPM) (Ho et al., 2020) and Score Matching with Langevin Dynamics (SMLD) , a subset of general score-based generative models (SGMs), (Song & Ermon, 2019), have achieved state-of-the-art quality in generating images, molecules, audio and graphs[1] (Song et al., 2021). Unlike GANs, training diffusion models is usually very stable and straightforward, they do not suffer as much from mode collapse issues, and they can generate images of similar quality.

In parallel with the success of these diffusion models, Song et al. (2021) demonstrated that both SGMs and DDPMs can mathematically be understood as variants of the same process. In both cases, the data distribution is progressively perturbed by a noise diffusion process defined by a specific Stochastic Differential Equation (SDE), which can then be time-reversed to generate realistic data samples from initial noise samples.

While the success of diffusion models has mainly been driven by data with Euclidean geometry (e.g., images), there is great interest in extending these methods to manifold-valued data, which are ubiquitous in many scientific disciplines. Examples include high-energy physics (Brehmer & Cranmer, 2020; Craven et al., 2022), astrophysics (Hemmati et al., 2019), geoscience (Gaddes et al., 2019), and biochemistry (Zelesko et al., 2020). Very recently, pioneering work has started to develop generic frameworks for defining SGMs on arbitrary compact Riemannian manifolds (De Bortoli et al., 2022), and non-compact Riemannian manifolds (Huang et al., 2022).

In this work, instead of considering generic Riemannian manifolds, we are specifically concerned with the Special Orthogonal group in 3 dimensions, SO(3), which corresponds to the Lie group of 3D rotations. Modeling 3D orientations is of particularly high interest in many fields including for instance in robotics (estimating the pose of an object, Hoque et al. 2021); and in biochemistry (finding the conformation angle of molecules that minimizes the binding energy, Mansimov et al. 2019). Contrary to more generic Riemannian manifolds, SO(3) benefits from specific properties, including a tractable heat kernel and efficient geometric ODE/SDE solvers, that will allow us to define very efficient diffusion models specifically for this manifold.

---

[1]For a comprehensive list of articles on score-based generative modeling, see `https://scorebasedgenerativemodeling.github.io/`

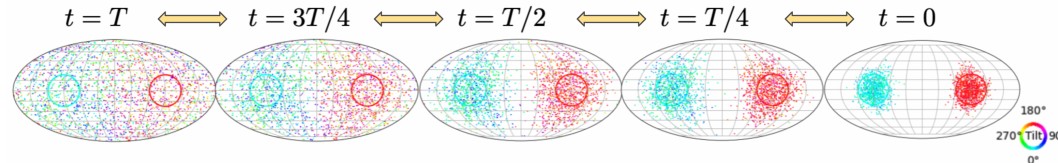

Figure 1: Illustration of reversible diffusion of a mixture of two $\mathcal{IG}_{\mathrm{SO}(3)}$ blobs on SO(3). Samples from a given base distribution (right most, denoted by circles) can be evolved under the probability flow ODE (Eq. 3) towards a noisy distribution (left most), or vice-versa from the noisy distribution back to the target distribution. Each point represents a rotation matrix in SO(3) projected on the sphere according to its canonical axis, the color indicates the tilt around that axis (visualisation adopted from Murphy et al. 2021). An animation of this figure is available at this link.

The contributions of our paper are summarized as follows:

- We reformulate Euclidean diffusion models on the SO(3) manifold, and demonstrate how the tractable heat kernel solution on SO(3) can be used to recover simple and efficient algorithms on this manifold.

- We provide concrete implementations of both Score-Based Generative Model and Denoising Diffusion Probabilistic Models specialized for SO(3).

- We reach a new state-of-the-art in sample quality on synthetic SO(3) distributions with our proposed SO(3) Score-Based Generative Model.

## 2 PRELIMINARIES AND NOTATIONS

In this work, we are exclusively considering the SO(3) manifold, corresponding to the Lie group of 3D rotation matrices. We will denote by $\exp : \mathfrak{so}(3) \rightarrow \mathrm{SO}(3)$ and $\log : \mathrm{SO}(3) \rightarrow \mathfrak{so}(3)$ the exponential and logarithmic maps that connect SO(3) to its tangent space and Lie algebra $\mathfrak{so}(3)$. $\mathfrak{so}(3)$ corresponds to all skew-symmetric 3x3 matrices in $\mathbb{R}^3$, which can be parameterised in terms of a vector in $\mathbb{R}^3$, which corresponds in turn to the *axis-angle* representation of rotation matrices. For any rotation $\mathbf{x} \in \mathrm{SO}(3)$, its axis-angle representation $\boldsymbol{\omega} \in \mathbb{R}^3$ can be computed as $\boldsymbol{\omega} = \omega \mathbf{v}$ with $\omega = \arccos\left(2^{-1}(\mathrm{tr}(\mathbf{x}) - 1)\right) \in (-\pi, \pi]$ and $\mathbf{v} = \frac{1}{2\sin\omega}(\mathbf{x}_{32} - \mathbf{x}_{23}, \mathbf{x}_{13} - \mathbf{x}_{31}, \mathbf{x}_{21} - \mathbf{x}_{12})$ a unit vector of $\mathbb{R}^3$.

We direct the interested reader to more information on representations of SO(3) in Appendix C.

## 3 DIFFUSION PROCESS ON SO(3)

Similarly to Euclidean diffusion models (Song et al., 2021), we begin by defining a Brownian noising process that will be used to perturb the data distribution. Let us assume a Stochastic Differential Equation of the following form:

$$\mathrm{d}\mathbf{x} = \mathbf{f}(\mathbf{x}, t)\,\mathrm{d}t + g(t)\,\mathrm{d}\mathbf{w}, \tag{1}$$

where $\mathbf{w}$ is a Brownian process on SO(3), $\mathbf{f}(\cdot\,, t) : \mathrm{SO}(3) \rightarrow T_{\mathbf{x}}\mathrm{SO}(3)$ is a drift term, and $g(\cdot) : \mathbb{R} \rightarrow \mathbb{R}$ is a diffusion term. If we sample initial conditions for this SDE at $t = 0$ from a given data distribution $\mathbf{x}(0) \sim p_{\mathrm{data}}$, we will denote by $p_t$ the marginal distribution of $\mathbf{x}(t)$ at time $t > 0$. Thus $p_0 = p_{\mathrm{data}}$, and at final time $T$ at which we stop the diffusion process $p_T$ will typically tend to a known target distribution that will be easy to sample from.

Just like in the Euclidean case, as demonstrated in De Bortoli et al. (2022), under mild regularity conditions Equation 1 admits a reverse diffusion process on compact Riemannian manifolds such as SO(3), defined by the following reverse-time SDE:

$$\mathrm{d}\mathbf{x} = [\mathbf{f}(\mathbf{x}, t) - g(t)^2 \nabla \log p_t(\mathbf{x})]\mathrm{d}t + g(t)\mathrm{d}\bar{\mathbf{w}}, \tag{2}$$

where $\bar{\mathbf{w}}$ is a reversed-time Brownian motion and the *score function* $\nabla \log p_t(\mathbf{x}) \in T_{\mathbf{x}}\mathrm{SO}(3)$ is the derivative of the log marginal density of the forward process at time $t$. Corresponding to this reverse-time SDE, one can also define a probability flow ODE (Song et al., 2021):

$$\mathrm{d}\mathbf{x} = [\mathbf{f}(\mathbf{x}, t) - g(t)^2 \nabla \log p_t(\mathbf{x})]\mathrm{d}t. \tag{3}$$

This deterministic process is entirely defined once the score is known and maps $p_T$ to any intermediate marginal distributions $\{p_t\}_{0 \le t < T}$ of the forward process, including $p_0$. In particular, it can be seen as the equivalent of Neural ODE-based Continuous Normalizing Flows (CNF, Chen et al., 2018) with an explicit parameterization in terms of the score function. We illustrate this process in Figure 1 with samples from two Gaussian-like blobs on SO(3) being transported reversibly through this ODE between $t = 0$ and $t = T$.

While these equations are direct analog of the Euclidean SDEs and ODE described in Song et al. (2021), defining diffusion generative models on SO(3) will mainly differ on the two following points:

- Defining the equivalent of the Gaussian heat kernel on SO(3): this is needed to easily sample from any intermediate $p_t$ without having to simulate an SDE.
- Solving SDEs and ODEs on the manifold: contrary to the Euclidean case, the diffusion process must remained confined to the SO(3) manifold, which requires specific solvers.

We address these two points below before moving on to defining our generative models on SO(3).

### 3.1 THE ISOTROPIC GAUSSIAN DISTRIBUTION ON SO(3)

In general, the main disadvantage of working on Riemannian manifolds compared to Euclidean space is that they lack a closed form expression for the heat kernel, i.e., the solution of the diffusion process (which is a Gaussian in Euclidean space). For compact manifolds, the heat kernel is in general only available as an infinite series, which in the case of SO(3), takes the following form (Nikolayev & Savyolov, 1970):

$$f_\epsilon(\omega) = \sum_{\ell=0}^{\infty} (2\ell + 1) \exp(-l(l+1)\epsilon^2) \frac{\sin((\ell + 1/2)\omega)}{\sin(\omega/2)} \tag{4}$$

where $\omega = |\boldsymbol{\omega}| \in (-\pi, \pi]$ is the rotation angle of the axis-angle representation $\boldsymbol{\omega}$ of a given rotation matrix and $\epsilon$ is a concentration parameter.

While for $\epsilon > 1$ this series converges quickly ($\ell_{\max} = 5$ is sufficient to achieve sub-percent accuracy), the convergence gets slower as $\epsilon$ gets smaller, which makes it impractical to model concentrated distributions. Thankfully, this series has been thoroughly studied in the literature and Matthies et al. (1988) shows that an excellent approximation of Equation 4 can be achieved for $\epsilon < 1$ using the following closed-form expression:

$$f_\epsilon(\omega) \simeq \sqrt{\pi} \epsilon^{-3/2} e^{\frac{\epsilon}{4} - \frac{(\omega/2)^2}{\epsilon}} \left( \frac{\omega - e^{-\frac{\pi^2}{\epsilon}} \left((\omega - 2\pi)e^{\pi\omega/\epsilon} + (\omega + 2\pi)e^{-\pi\omega/\epsilon}\right)}{2\sin(\omega/2)} \right). \tag{5}$$

Therefore, in practical applications, one can switch between using a truncation of Equation 4 for $\epsilon \ge 1$ and the approximation Equation 5 for $\epsilon < 1$.

Because of the property of being a solution of a diffusion process on SO(3), $f_\epsilon$ can be used to define the manifold equivalent of the Euclidean isotropic Gaussian distribution, which we will refer to as $\mathcal{IG}_{\mathrm{SO}(3)}$, the Isotropic Gaussian on SO(3) (Leach et al., 2022; Ryu et al., 2022), also known in the literature as the normal distribution on SO(3) (Nikolayev & Savyolov, 1970; Matthies et al., 1988). For a given mean rotation $\boldsymbol{\mu} \in \mathrm{SO}(3)$ and scale $\epsilon$, the probability density of a rotation $\mathbf{x} \in \mathrm{SO}(3)$ under $\mathcal{IG}_{\mathrm{SO}(3)}(\boldsymbol{\mu}, \epsilon)$ is given by:

$$\mathcal{IG}_{\mathrm{SO}(3)}(\mathbf{x}; \boldsymbol{\mu}, \epsilon) = f_\epsilon(\arccos\left[2^{-1}(\mathrm{tr}(\boldsymbol{\mu}^T\mathbf{x}) - 1)\right]). \tag{6}$$

Sampling from $\mathcal{IG}_{\mathrm{SO}(3)}(\boldsymbol{\mu}, \epsilon)$ is achieved in practice by inverse transform sampling. The cumulative distribution function over angles needed to sample with respect to the uniform distribution on SO(3) can be evaluated numerically given integrating $\frac{1 - \cos(\omega)}{\pi} f_\epsilon(\omega)$ over $(-\pi, \pi]$. To form a rotation matrix $\mathbf{x} \sim \mathcal{IG}_{\mathrm{SO}(3)}(\cdot; \boldsymbol{\mu}, \epsilon)$, one therefore first samples a rotation angle by inverse transform sampling

---

**Algorithm 1** Geometric ODE solver on SO(3) (Heun's method) for $\mathrm{d}\mathbf{x} = \mathbf{f}(\mathbf{x}, t)\,\mathrm{d}t$

---

**Require:** Step size $h$, initial condition $\mathbf{x}_0$, time steps $\{t_n\}_{n=0}^N$, number of steps $N$
1: **for** $n \in \{0, \ldots, N-1\}$ **do**
2:    $\mathbf{y}_1 = h\,\mathbf{f}(\mathbf{x}_n,\, t_n)$
3:    $\mathbf{y}_2 = h\,\mathbf{f}(\exp(\frac{1}{2}\mathbf{y}_1)\mathbf{x}_n,\, t_n + \frac{1}{2}h)$
4:    $\mathbf{x}_{n+1} = \exp(\mathbf{y}_2)\mathbf{x}_n$
5: **end for**
6: **return** $\{\mathbf{x}_n\}_{n=0}^N$

---

given this CDF, then samples uniformly on $\mathcal{S}^2$ a rotation axis $\mathbf{v}$, yielding an axis-angle representation of a rotation matrix $\boldsymbol{\omega} = \omega\mathbf{v}$, which is then shifted by the mean of the distribution according to $\boldsymbol{x} = \boldsymbol{\mu}\exp(\boldsymbol{\omega})$.

An important property of $\mathcal{IG}_{\text{SO(3)}}(\mu, \epsilon)$, which sets it apart from other distributions on SO(3) (e.g. Bingham, Matrix Fisher, Wrapped Normal, more on this in Appendix E.1 ), is that it remains *closed under convolution*, as a direct consequence of being the solution of a diffusion process. The convolution of two centered $\mathcal{IG}_{\text{SO(3)}}$ distributions of scale parameter $\epsilon_1$ and $\epsilon_2$ is an $\mathcal{IG}_{\text{SO(3)}}$ distribution of scale $\epsilon_1 + \epsilon_2$.

We will also note two interesting asymptotic behaviors. For large $\epsilon$, it tends to $\mathcal{U}_{\text{SO(3)}}$, the uniform distribution on SO(3), while for small $\epsilon$ the distribution $\mathcal{IG}_{\text{SO(3)}}(\mathbf{I}, \epsilon)$ can locally be approximated in the axis-angle representation of the tangent space by a normal distribution $\mathcal{N}(0, \sigma^2\mathbf{I})$ in $\mathbb{R}^3$, with $\epsilon = \frac{\sigma^2}{2}$.

### 3.2 SOLVING ORDINARY DIFFERENTIAL EQUATIONS ON SO(3)

Thanks to the existence of a tractable heat kernel on SO(3), the generative models we will define in the next section will not actually require us to solve the SDEs introduced at the beginning of this section, and we will only need to solve the probability flow ODE defined in Equation 3.

Solving differential equations on manifolds can broadly be achieved using two distinct strategies, either projection methods using a Euclidean solver followed by a projection step onto the manifold, or intrinsic methods that rely on additional structure of the manifold to define an iteration that remains by construction on the manifold. In this work, we are concerned with SO(3), which is not only a compact Riemannian manifold, but also possesses a Lie group structure, which makes it amenable to efficient solvers. In particular, we will make use of the Runge-Kutta-Munthe-Kaas (RK-MK) class of algorithms and direct the interested reader to Iserles et al. (2000) for a review of Lie group integrators. We adopt in practice the Lie group equivalent of Heun's method, which is one variant of RK-MK integrators, and we provide the details of this integrator in Algorithm 1.

While we will not require it in practice, it is also possible to build SDE solvers on SO(3) with a similar strategy, and we point the interested reader for instance to the Geodesic Random Walk algorithm described in De Bortoli et al. (2022).

## 4 DIFFUSION GENERATIVE MODELS ON SO(3)

The core idea of diffusion models is to perturb a given empirical data distribution $p_{\text{data}}$ by a noise process defined in terms of a stochastic differential equation of the form Equation 1. While several expressions can be proposed for this SDE, for simplicity we will consider here the case of the Variance-Exploding SDE (for our experiments we use a Variance-Preserving SDE in the DDPM case), with $\mathbf{f}(\mathbf{x}, \mathbf{t}) = 0$ and $g(t) = \sqrt{\frac{\mathrm{d}\epsilon(t)}{\mathrm{d}t}}$ for a given choice of noise schedule $\epsilon(t)$, which corresponds to the canonical choice of the Euclidean Score-Matching Langevin Dynamics (Song & Ermon, 2019):

$$\mathrm{d}\mathbf{x} = \sqrt{\frac{\mathrm{d}\epsilon(t)}{\mathrm{d}t}}\,\mathrm{d}\mathbf{w}\,. \tag{7}$$

For our fiducial model, and unless stated otherwise, we will further assume for simplicity the following noise schedule: $\epsilon(t) = t$. The main drawback of this SDE in Euclidean geometry is that it



Figure 2: Sampling from a Diffusion Generative Model trained on a synthetic density on SO(3). Starting from $\mathcal{U}_{SO(3)}$, the uniform distribution on SO(3) at $t = T$ (left), the sampling procedure (either based on SGMs or DDPMs) denoises this distribution back to the target density at $t = 0$ (right). For visualization this density plot shows the distribution of canonical axes of sampled rotations projected on the sphere; the tilt around that axis is discarded.

will tend to a Gaussian with infinitely large variance. However, on SO(3) this SDE will tend to the uniform distribution $\mathcal{U}_{SO(3)}$ which is a natural choice for the prior distribution at large $T$.

Following from this choice of SDE, we can define a noise kernel $p_\epsilon(\tilde{\boldsymbol{x}}|\boldsymbol{x}) = \mathcal{IG}_{SO(3)}(\tilde{\boldsymbol{x}}; \boldsymbol{x}, \epsilon)$ for $\boldsymbol{x}, \tilde{\boldsymbol{x}} \in$SO(3), such that the data distribution convolved by this noise kernel becomes

$$p_\epsilon(\boldsymbol{x}) = \int_{SO(3)} p_{\text{data}}(\boldsymbol{x}')p_\epsilon(\boldsymbol{x}|\tilde{\boldsymbol{x}})\,\mathrm{d}\boldsymbol{x} \ , \tag{8}$$

and corresponds to $p_t$, the marginal distribution of the diffusion process at time $t$: $p_{\epsilon(t)} = p_t$.

Having introduced a specific choice of kernel and SDE well suited to the SO(3) manifold, we now move on to describing the two different approaches to build generative models: Score-Based Models and Denoising Diffusion Probabilistic Models. They both will lead to sampling procedures illustrated in Figure 2.

### 4.1 Score-based Generative Model

The first strategy directly extends Euclidean SGMs (Song & Ermon, 2019; Song et al., 2021) and relies on the time-reversed diffusion process described in Equation 2. Samples from the learned distribution $p_0$ can be sampled by first sampling $\mathbf{x}_T \sim \mathcal{U}_{SO(3)}$ and evolving these samples either through the reverse SDE (Equation 2) or probability flow ODE (Equation 3) back to $t = 0$. This process is entirely defined as soon as the *score function* of the marginal distribution at any intermediate time $t$, $\nabla \log p_{\epsilon(t)}$, is known. Therefore the first step is to establish a score-matching strategy on SO(3).

Let us consider $\{X_i\}_{i=0}^3$, an orthonormal basis of the tangent space $T_{\mathbf{e}}$SO(3). The directional derivative of the log density of the noise kernel $p_\epsilon(\boldsymbol{x}|\tilde{\boldsymbol{x}})$ can be computed as:

$$\nabla_{X_i} \log p_\epsilon(\tilde{\mathbf{x}}|\mathbf{x}) = \frac{\mathrm{d}}{\mathrm{d}s} \log p_\epsilon(\tilde{\mathbf{x}} \exp(sX_i)|\mathbf{x})\bigg|_{s=0} \ , \tag{9}$$

which can be computed in practice by automatic differentiation given the explicit approximation formulae for the $\mathcal{IG}_{SO(3)}$ distribution introduced in subsection 3.1. To match this derivative, we introduce a neural score estimator $s_\theta(\mathbf{x}, \epsilon) : SO(3) \times \mathbb{R}^{+\star} \to \mathbb{R}^3$, which can be trained directly under a conventional denoising score matching loss:

$$\mathcal{L}_{DSM} = \mathbb{E}_{p_{\text{data}}(\mathbf{x})}\mathbb{E}_{\epsilon \sim \mathcal{N}(0, \sigma_\epsilon^2)}\mathbb{E}_{p_{|\epsilon|}(\tilde{\mathbf{x}}|\mathbf{x})}\left[|\epsilon| \ \| s_\theta(\tilde{\mathbf{x}}, \epsilon) - \nabla_X \log p_{|\epsilon|}(\tilde{\mathbf{x}}|\mathbf{x}) \|_2^2\right] \tag{10}$$

where we sample at training time random noise scales $\epsilon \sim \mathcal{N}(0, \sigma_\epsilon^2)$ similarly to Song & Ermon (2020). The minimum of this loss will be achieved for $s_\theta(\mathbf{x}, \epsilon) = \nabla \log p_\epsilon$.

Once the score function is estimated from data using this score matching loss, sampling from the generative model can be achieved by using the reverse SDE formula, or using the ODE flow formula. In this work, we use the latter for its simplicity and speed, so that our specific fiducial sampling strategy becomes:

$$\mathbf{x}_T \sim \mathcal{U}_{SO(3)} \qquad ; \qquad \mathrm{d}\mathbf{x}_t = -\frac{1}{2}\frac{\mathrm{d}\epsilon(t)}{\mathrm{d}t} s_\theta(\mathbf{x}_t, \epsilon(t))\,\mathrm{d}t \tag{11}$$

which we solve down to $t = 0$ with the geometric ODE solver described in Algorithm 1. Compared to stochastic sampling strategies based on simulating the reverse SDE, this approach has several

---

**Algorithm 2** Sampling from Denoising Diffusion Probabilistic Model on SO(3)

---

**Require:** Trained neural networks $\mu_\theta(\mathbf{x}, t), \epsilon_\theta(\mathbf{x}, t)$, number of steps $N$, time steps $\{t_i\}_{i=0}^N$
1: $\mathbf{x}_N \sim \mathcal{U}_{\mathrm{SO}(3)}$
2: **for** $i = \{N, N-1, \ldots, 1\}$ **do**
3: $\quad \mathbf{x}_{i-1} \sim p_\theta(\cdot; \mathbf{x}_i) = \mathcal{IG}_{\mathrm{SO}(3)}(\cdot; \exp(\mu_\theta(\mathbf{x}_i, t_i)), \epsilon_\theta(\mathbf{x}_i, t_i))$
4: **end for**
5: **return** $\{\mathbf{x}_n\}_{n=0}^N$

---

advantages. 1) It is much faster, and can benefit from adaptive ODE solvers bringing down the number of score evaluations needed, 2) the same ODE can be used to evaluate the log likelihood of the model by applying the probability flow formula of CNFs.

## 4.2 DENOISING DIFFUSION PROBABILISTIC MODEL

As described in Song et al. (2021), when using a finite number of steps, the forward diffusion process defined by Equation 7 $\{\mathbf{x}_i\}_{i=0}^N$ (corresponding to times $\{0 \leq t_i \leq T\}_{i=0}^n$) can be interpreted as a Markov process:

$$p(\mathbf{x}_{0:N}) = p(\mathbf{x}_0)p_{\epsilon_1}(\mathbf{x}_1|\mathbf{x}_0)\ldots p_{\epsilon_2}(\mathbf{x}_i|\mathbf{x}_{i-1})\ldots p_{\epsilon_N}(\mathbf{x}_N|\mathbf{x}_{N-1}) \tag{12}$$

with the transition kernel $p_{\epsilon_{i+1}}(\mathbf{x}_{i+1}|\mathbf{x}_i) = \mathcal{IG}_{\mathrm{SO}(3)}(\mathbf{x}_{i+1}; \mathbf{x}_i, \epsilon_{i+1})$, where $\epsilon_{i+1} = \epsilon(t_{i+1}) - \epsilon(t_i)$.

The idea of DDPMs is to introduce a reverse Markov process defined in terms of variational transition kernels $p_\theta(\boldsymbol{x}_{i-1}|\boldsymbol{x}_i)$:

$$p_\theta(\mathbf{x}_{0:N}) = p_\theta(\boldsymbol{x}_N)p_\theta(\mathbf{x}_{N-1}|\mathbf{x}_N)\ldots p_\theta(\mathbf{x}_{i-1}|\mathbf{x}_i)\ldots p_\theta(\mathbf{x}_0|\mathbf{x}_1). \tag{13}$$

While one could choose any distribution on SO(3) to parameterize this inverse transition kernel (e.g., Matrix Fisher, Bingham), we adopt for convenience an Isotropic Gaussian on SO(3) and use the following expression:

$$p_\theta(\boldsymbol{x}_{i-1}|\boldsymbol{x}_i) = \mathcal{IG}_{\mathrm{SO}(3)}(\boldsymbol{x}_{i-1}; \boldsymbol{x}_i \, \boldsymbol{\delta}_\theta(\boldsymbol{x}_i, t_i), \epsilon_\theta(\boldsymbol{x}_i, t_i)) \tag{14}$$

where $\boldsymbol{\delta}_\theta : \mathrm{SO}(3) \times \mathbb{R}^+ \to \mathrm{SO}(3)$ is a neural network predicting the residual rotation to apply to $\boldsymbol{x}_i$ to obtain the mean of the reverse kernel and $\epsilon_\theta : \mathrm{SO}(3) \times \mathbb{R}^+ \to \mathbb{R}^+$ is a neural network predicting the variance of this reverse kernel. To parameterize the output of $\boldsymbol{\delta}_\theta$ we adopt the 6D continuous rotation representation of (Zhou et al., 2019) and explore the impact of this choice in Appendix D.

If the reverse Markov process can be successfully trained to match the forward process, it provides a direct sampling strategy to generate samples from $p_0$ by initializing the chain from $p_T$ and iteratively sampling from the reverse kernel $p_\theta(\boldsymbol{x}_{i-1}|\boldsymbol{x}_i)$.

In DDPMs, the training strategy is to write down the Evidence Lower Bound (ELBO), given this variational approximation for the reverse Markov process, in order to train the individual transition kernels $p_\theta(\mathbf{x}_{i-1}|\mathbf{x}_i)$. To reduce the variance of this loss over a naive evaluation of the ELBO, Sohl-Dickstein et al. (2015) and Ho et al. (2020) propose to use a closed form expression of the reverse kernel $p(\mathbf{x}_{i-1}|\mathbf{x}_i, \mathbf{x}_0)$ when conditioned on $\mathbf{x}_0$. This makes it possible to rewrite the ELBO in terms of analytic KL divergences between Gaussian transitions kernels. However, contrary to the Gaussian case of Euclidean DDPMs, for $\mathcal{IG}_{\mathrm{SO}(3)}$ we do not easily have access to a closed form expression of the reverse kernel $p(x_{t-1}|x_t, x_0)$ which is needed to derive the training loss used in Ho et al. (2020). The same approach cannot be applied.

Instead, we consider the expression for the ELBO:

$$\mathbb{E}\left[-\log p_\theta(\mathbf{x}_0)\right] \leq \mathbb{E}_p\left[-\log p(\mathbf{x}_N) - \sum_{i \geq 1} \log \frac{p_\theta(\mathbf{x}_{i-1}|\mathbf{x}_i)}{p(\mathbf{x}_i|\mathbf{x}_{i-1})}\right] =: \mathcal{L}_{\mathrm{ELBO}} \tag{15}$$

which will be optimized by maximizing the log likelihood of individual transition kernels $\log p_\theta(\mathbf{x}_{i-1}|\mathbf{x}_i)$ over samples $\mathbf{x}_{i-1}, \mathbf{x}_i$ obtained through simulating the forward Markov diffusion process over the training set. Our strategy on SO(3), is therefore to train each transition kernel by maximum likelihood using the following loss function:

$$\mathcal{L}_{DDPM} := \sum_{i \geq 0} \mathbb{E}_{p_{\mathrm{data}}(\mathbf{x}_0)} \mathbb{E}_{p_\epsilon(\mathbf{x}_i|\mathbf{x}_0)} \mathbb{E}_{p_{\epsilon_i}(\mathbf{x}_{i+1}|\mathbf{x}_i)}\left[-\log p_\theta(\mathbf{x}_i|\mathbf{x}_{i+1})\right] \tag{16}$$

where the log probability of the $\mathcal{IG}_{\mathrm{SO}(3)}$ distribution used in our parameterised reverse kernel is defined in Equation 6. While this loss can indeed be used to train a DDPM (as demonstrated in the next section), compared to the strategy of Ho et al. (2020), we expect it to suffer from larger variance and is not explicitly parameterised in terms of the score function (Song et al., 2021). Once trained, we can use the sampling strategy described in Algorithm 2 to draw from the generative model.

## 5 RELATED WORK

Most related to our work is Song et al. (2021) which introduces the diffusion framework we use in this paper, and served as a point of reference throughout. We survey below related works that have developed methodologies to represent distributions on SO(3).

**Directional statistics**  The classical approach for modeling distributions on SO(3) relies on (mixtures) of analytic distributions defined over the group of rotations. Common examples of using such distributions for modeling uncertainties over orientations include the Bingham distribution (Peretroukhin et al., 2020; Srivatsan et al., 2018a; Gilitschenski et al., 2020) or the matrix Fisher distribution (Mohlin et al., 2020). The two main issues of these approaches are the lack of flexibility/expressivity of these analytic distributions, and the general difficulty of computing their normalization constant, which is typically required to train these models by maximum likelihood.

**Normalizing Flows**  A number of approach have been proposed to build density estimators on manifolds (which include SO(3)) based on Normalizing Flows. A first class of methods proposes to use a conventional Euclidean Normalizing Flow in $\mathbb{R}^n$, which is then mapped to the target manifold using an invertible mapping (Gemici et al., 2016; Falorsi et al., 2019). This has some limitations however as the target manifold needs to be homeomorphic to $\mathbb{R}^n$ (which is the case for SO(3)), and this mapping can also present discontinuities. As an improvement over this approach, a second class of methods based on continuous normalizing flows (Chen et al., 2018) has emerged, defining directly flows on the manifold (Falorsi & Forré, 2020; Mathieu & Nickel, 2020). These approaches remain relatively costly as training requires backpropagating through an ODE solver. Rozen et al. (2021) proposes to sidestep that issue by training the CNF through penalizing the divergence of the neural network. And finally, in recent work (Ben-Hamu et al., 2022) proposes to train a flow on manifolds by penalizing a Probability Path Divergence (PPD).

**Diffusion models**  In concurrent work, Leach et al. (2022) proposed an implementation of DDPMs on SO(3) by direct analogy with Ho et al. (2020), based on the Isotropic Gaussian on SO(3) as a replacement for the Normal distribution in $\mathbb{R}^n$. However, as mentioned in the previous section, the loss function used in Euclidan DDPMs does not directly translate to SO(3), which leads to imperfect density estimation as we will illustrate in our experiments. Finally, De Bortoli et al. (2022); Huang et al. (2022); Thornton et al. (2022) introduce generic frameworks for diffusion models on Riemannian manifolds but only for Score-Based Generative Model (SGM). Their generic approach means they do not benefit from the knowledge of a solution to the heat equation in SO(3), which we use extensively in our work to avoid the need to simulate SDEs and to efficiently generate samples from the forward diffusion process. In addition, we note that the method developed in Huang et al. (2022) is not particularly efficient on the orthogonal group as it requires a projection operation, which involves a singular value decomposition.

**Other approaches**  Murphy et al. (2021) develops a non-parametric representation of distributions on SO(3) by introducing a neural network to implicitly represent an unnormalized density on SO(3). Training this model by maximum likelihood requires computing the normalization constant of this implicit probability density function through brute-force evaluation on a tiling of SO(3), which is very costly in memory and limits the effective resolution of the learned densities.

## 6 EXPERIMENTS

We investigate the quality of the generative models described in the previous section on a series of synthetic test densities on SO(3). Details of the training procedures and architectural choices for all models can be found in Appendix A.

| Model | Checkerboard | 4-Gaussians | 3-Stripes |
|---|---|---|---|
| SGM on SO(3) (ours) | **0.50**$\pm$ 0.01 | **0.50**$\pm$ 0.01 | **0.51**$\pm$ 0.01 |
| DDPM on SO(3) (ours) | 0.52$\pm$ 0.01 | 0.53$\pm$ 0.01 | 0.52$\pm$ 0.01 |
| RSGM (De Bortoli et al., 2022) | 0.51$\pm$ 0.01 | – | **0.51** $\pm$ 0.01 |
| Moser Flow(Rozen et al., 2021) | 0.56$\pm$ 0.01 | 0.60$\pm$ 0.02 | 0.53$\pm$ 0.02 |
| DDPM (Leach et al., 2022) | 0.71$\pm$ 0.04 | 0.90$\pm$ 0.05 | 0.60$\pm$ 0.03 |
| Implicit-PDF (Murphy et al., 2021) | 0.59$\pm$ 0.04 | 0.81$\pm$ 0.09 | 0.63$\pm$ 0.04 |

Table 1: Sample quality metric from the C2ST (lower is better). If the learned distribution is identical to the original one, the metric should be $\sim 0.5$; if it is significantly different, the metric tends towards $\sim 1$. The errors on the metric were obtain from the standard deviation of the metric over $k$-fold cross validation samples for a single training of the model. – indicate a failure to evaluate the metric for a particular model.

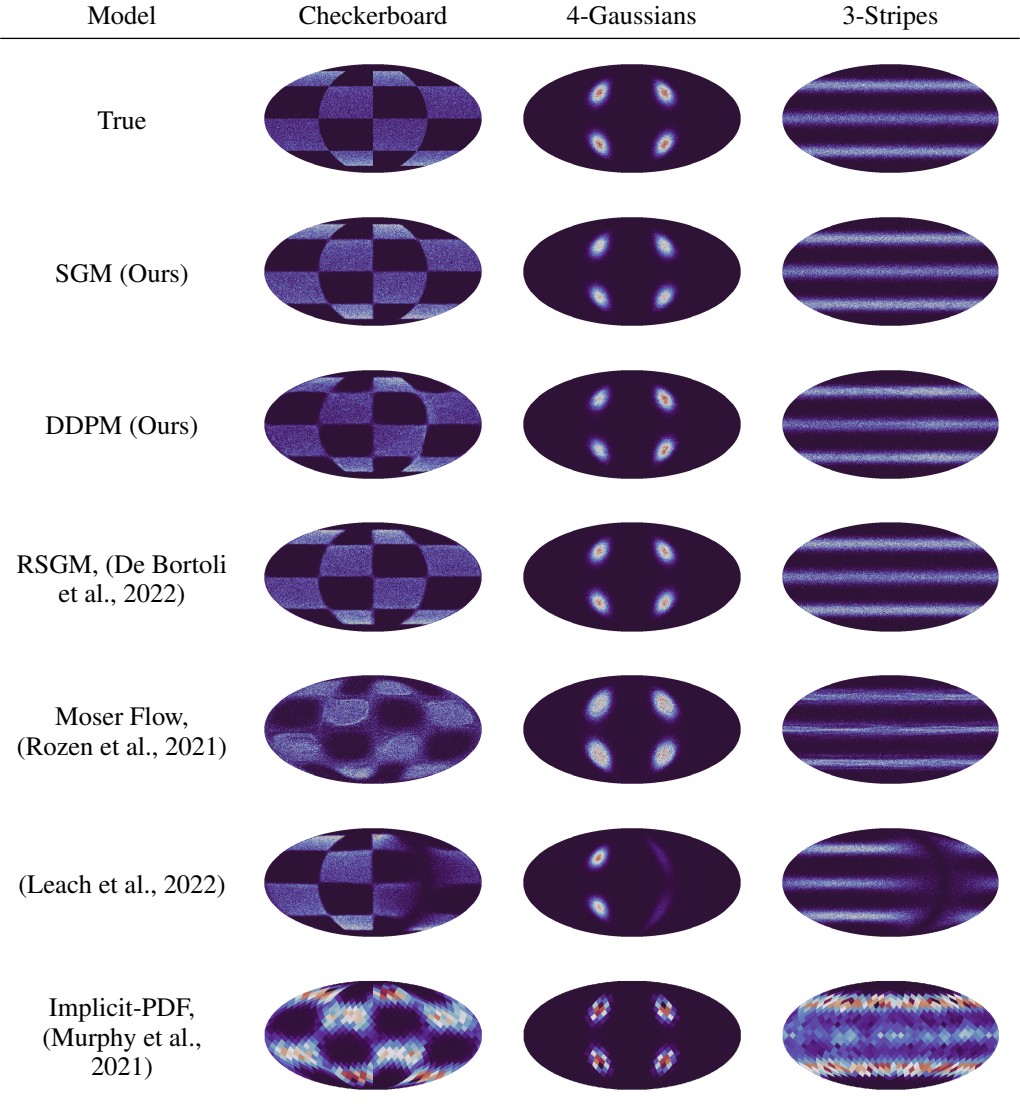

Figure 3: Density plot comparing samples from learned synthetic densities on SO(3). For visualization this density plot shows the distribution of canonical axes of sampled rotations projected on the sphere; the tilt around that axis is discarded.

| Image | True | Scatter (Predicted) | Density (Predicted) |
|-------|------|--------------------|--------------------|

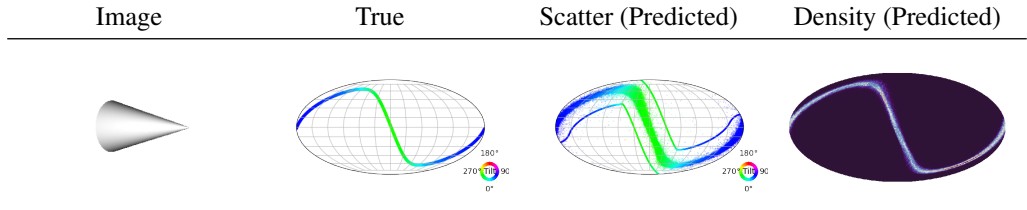

Figure 4: Example of pose estimation using an SO(3) SGM conditioned on images from the SYMSOL dataset Murphy et al. (2021).

**Test densities on SO(3)** We adopt three different toy distributions on SO(3): a checkerboard pattern, a multi-modal distribution of 4 concentrated Gaussians and a stripe pattern that can be viewed as circles on the sphere. We focus on evaluating the generative models in terms of the quality of their sample generation using the Classifier 2-Sample Tests (C2ST) metric (Lopez-Paz & Oquab, 2017; Dalmasso et al., 2020; Lueckmann et al., 2021). The C2ST metric has been used in particular in the context of simulation-based inference to quantify the quality of inferred distributions. Concisely, the C2ST method uses a neural network classifier to discriminate between true and the generated samples, yielding a value of $0.5$ if the two distributions are perfectly indistinguishable to the classifier, up to a value of $1$ if they are extremely different. In contrast to the usual Negative Log Likelihood (NLL), C2ST can be consistently computed for all generative models we compare bellow.

We present in Figure 3 and Table 1 the results of our (SGM, DDPM-VExp, DDPM-VPres) comparisons on these test densities against the implicit-pdf method of Murphy et al. (2021), the DDPM implementation of Leach et al. (2022), Moser flow of Rozen et al. (2021), and the Riemannian Score-Based Generative Model (RSGM) of De Bortoli et al. (2022) (trained under their $\ell_{t|0}$ score matching loss). We find that in all cases our SGM implementation on SO(3) yields the best C2ST metric, which is in line with the visual quality of distributions shown in Figure 3. Our DDPM implementation on SO(3) yields distributions that are comparatively less sharp, which we attribute to the larger variance of our training loss for that model. Compared to other models, our experiments illustrate a failure mode in the method of Leach et al. (2022) which we attribute to the fact that the usual DDPM loss function cannot be directly translated to SO(3) (as discussed in subsection 4.2). We also note that the Implicit-PDF model, in comparison, is extremely limited in resolution because of the memory cost of evaluating the pdf on a tiling of SO(3), and thus yields much lower scores. The best results after our method are achieved by the RSGM model (De Bortoli et al., 2022), which is expected due to its similarity with our work, but is slower to train in the specific case of SO(3). We find that the cost of simulating the forward SDE in the training phase leads to a factor x8 in computation time per batch on a given GPU.

**Pose estimation** To test practical applications of our model, following (Murphy et al., 2021) we used a vision description obtained from a pre-trained ResNet architecture with ImageNet weights consisting of 2048 dimensional vector to condition an SO(3) SGM. Using images of symmetric solids from the SYMSOL dataset Murphy et al. (2021) we show that we can correctly estimate poses of objects with degenerate symmetry, as shown in Fig. 4. (and in Appendix B).

## 7 CONCLUSIONS AND DISCUSSION

In this paper, we have presented a framework for score-based diffusion generative models on SO(3), as an extension of Euclidean SDE-based models (Song et al., 2021). Because it is developed specifically for the SO(3) manifold, our work proposes a simpler and more efficient alternative to other recent (and general) Riemannian diffusion models while reaching state-of-the-art quality on synthetic distributions on SO(3). One of the most promising applications of this work is in robotics and computer vision, for the general task of pose-estimation, where our proposed model significantly outperforms current baselines (Murphy et al., 2021). In the natural sciences, generative models on SO(3) are also of great interest and can be used for instance to find the angle of a molecule that minimizes the binding energy. Finally we note that as an interesting extension of the models presented in this work, one could define a Schrödinger bridge approach (De Bortoli et al., 2021; Thornton et al., 2022) specifically for SO(3), which would improve both sampling efficiency and sample quality.

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

## A  IMPLEMENTATION AND TRAINING

We designed our neural networks with a size of {256,256,256,256,256} neurons each with leaky ReLU activation and with a residual connection. The neural networks were implemented using the axis-angle representation of SO(3), i.e. the input and output elements were represented using axis-angle representation. Additionally, the neural networks were conditioned on the noise scheduler and the noise scales were also learned parameters. We trained our models using the Adam optimizer with learning rate of $10^{-4}$, exponential decay rates of $\beta_1 = 0.90$ and $\beta_2 = 0.95$, 400 000 iterations, and a batch size of 1024. NVIDIA Tesla V100 GPU was used as the hardware, with JAX and DeepMind-Haiku Python libraries as the software.

For the DDPM, we adopt in practice the Variance Preserving SDE of (Ho et al., 2020) as we obtain better results empirically than with a Variance Exploding SDE.

## B  ADDITIONAL POSE ESTIMATION RESULTS

Here we provide on Figure 5 additional results on pose estimation.

## C  REPRESENTATIONS OF SO(3)

The special orthogonal group, SO(3), is the Lie group of all rotations about the origin in 3-dimensional space. There are several ways to represent the elements of the group SO(3), each with its advantages and disadvantages:

- Rotation Matrices $\in \mathbb{R}^{3 \times 3}$ with determinant equal to 1. This representation has 9 parameters and can be subject to some numerical stabilities, such as when computing the inverse or trigonometric functions.

- Euler angles (also called *yaw, pitch, and roll* in robotics) are three angles $\alpha, \beta, \psi$ that can describe an orientation with respect to a fixed coordinate system. This representation is subject to the infamous Gimbal lock, where one degree of freedom is lost when two axes of the gimbal become parallel.

- Unit Quaternions defined as $\gamma = a + b\mathbf{i} + c\mathbf{j} + d\mathbf{k}$, where $a, b, c, d$ are real number satisfying $\sqrt{a^2 + b^2 + c^2 + d^2} = 1$ with $\mathbf{i} + \mathbf{j} + \mathbf{k}$ denoting the vector (or imaginary) part of the unit quaternion. This representation has 4 parameters and has elegant operations (Hamilton product) without trigonometric functions E. However, quaternions are antipodally symmetric which introduces some degeneracies.

- Axis - angle representation (normalized), Tangent space (unnormalized ) defined as $\theta = \theta \mathbf{e} = (\theta_1, \theta_2, \theta_3) = \theta(e_1, e_2, e_3)$, where $\theta$ is the rotation angle and $\mathbf{e}$ is the rotation axis. However, this representation does not have well defined operations to combine rotations, and is furthermore discontinuous at $\theta = \pi$ (Zhou et al., 2019).

Therefore, in practice it is best to use some combinations of the aforementioned representations and convert back and forth among them. For a comprehensive review on SO(3) representations and metrics, especially for computer scientists, please refer to (Hartley et al., 2013).

| Image | True | Scatter (Predicted) | Density (Predicted) |
|-------|------|---------------------|---------------------|

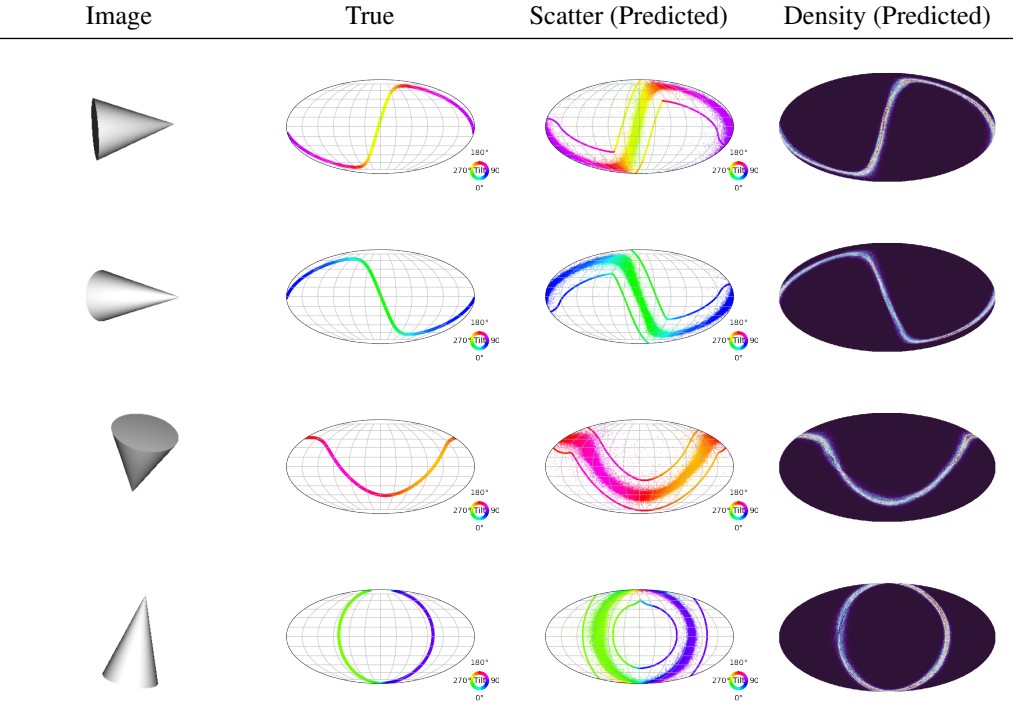

Figure 5: Predicted poses for an image of a solid with degenerate symmetry, here we only show it for the cone. The 1st column depicts the image of the symmetric solid. In column 2, each point represents a rotation matrix in SO(3) projected on the sphere according to its canonical axis, the color indicates the tilt around that axis. For visualization the density plot (column 3) shows the distribution of canonical axes of sampled rotations projected on the sphere; the tilt around that axis is discarded.

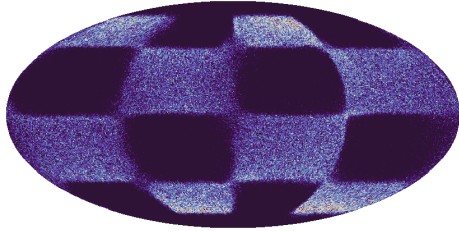
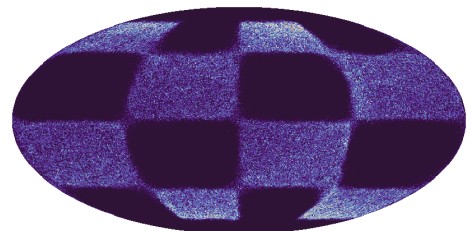
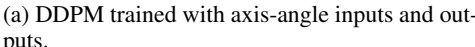

(a) DDPM trained with axis-angle inputs and out-
puts.

(b) DDPM trained with 3x3 rotation matrices as
inputs and using a continuous 6D output rotation
representation.

Figure 6: Comparison of distributions sampled from the DDPM under two different parameteriza-
tions of both input and output rotations. Once the models are fully trained as illustrated here, the
impact is small, but on a partially trained network discontinuities would be visible in the case of the
axis-angle representation, mostly due to the discontinuity in the input rotations.

## D  IMPACT OF ROTATION REPRESENTATIONS ON NEURAL DIFFUSION
## MODEL

As highlighted in Zhou et al. (2019), a particular choice of rotation representation can affect the
training and accuracy of neural networks which either take rotations as an input or that output rota-
tions. In particular, common representations such as axis-angle and quaternions are known to have
discontinuities, which are needlessly difficult to capture for a neural network. In that work, they
propose in particular to use 5 or 6 dimensional representations which have the particularity of being
continuous and demonstrate their benefit in neural network training.

In our work, we make the choice of providing as inputs of the neural networks directly the 3x3
rotation matrix. Only the network involved in the DDPM needs to represent rotations as an output,
and there we adopt the 6D representation following Zhou et al. (2019), which can be seen as two 3D
vectors, from which we can build a full orthogonal rotation matrix using cross-products.

In comparing the impact of this choice against using only an axis-angle representation as inputs and
outputs, we observe the following points:

- The choice of the output parameterization (in the case of the DDPM) has no noticeable
  effect, which is expected as the model outputs residual rotations, which remain small and
  thus away from the discontinuity in the axis-angle representation.

- While the differences are small once the networks are fully trained, as illustrated on Fig-
  ure 6, we notice for partially trained networks a discontinuity in the sampled distributions in
  the case of an input axis-angle representation. Therefore, we directly feed the 3x3 rotation
  matrix as an input to our networks.

## E  QUATERNION OPERATIONS FOR SO(3)

Quaternions form a group under multiplication, defined by the Hamilton product. Given quternions
$\gamma_1$ and $\gamma_2$, the Hamilton product is defined by carrying out the $\gamma_1 \cdot \gamma_2 = (a_1 + b_1\mathbf{i} + c_1\mathbf{j} + d_1\mathbf{k}) \cdot
(a_2 + b_2\mathbf{i} + c_2\mathbf{j} + d_2\mathbf{k})$ in a distributive manner, keeping in mind the basis multiplication identities.
This operation is physically equivalent to rotating by $\gamma_1$ and then by $\gamma_2$. The identity of the group
is the quaternion $\gamma_0 = 1 + 0\mathbf{i} + 0\mathbf{j} + 0\mathbf{k}$ and the inverse of $\gamma*$ (also conjugate) is defined as
$\gamma* = a - b\mathbf{i} - c\mathbf{j} - d\mathbf{k}$.

**The reparametrization trick and variance preserving quaternions**  In Euclidean space for a
Gaussian random variable $\mathbf{x}$ from $\mathcal{N}(\mu; \sigma^2\mathbf{I})$, the reparametrization trick is defined as $\mathbf{x} = \mu + \sigma^2 \cdot \delta$

where $\delta \sim \mathcal{N}(0; \mathbf{I})$. We can define a analogous operation in the quaternion group, as such:

$$\gamma = \theta \cdot \epsilon^\delta \tag{17}$$

Here, the quternion raised to some scalar power is defined as the $\gamma^a = \exp(\ln(\gamma)a)$, in other words we take the quaternion to the tangent space from the manifold, perform the operation of multiplication and bring it back into the manifold by using the exponential map. and the variance preserving operation analogous to $\sqrt{\alpha} \cdot \mathbf{x} + \sqrt{(1-\alpha)}\delta$ can be defined as

$$\mathbf{x} = \mathbf{x}^{\sqrt{\alpha}} \cdot \delta^{\sqrt{(1-\alpha)}} \tag{18}$$

For the variance exploding case, we can directly sample from the heat kernel without resorting to the quaternion operations.

### E.1 DISTRIBUTIONS ON SO(3)

In literature there are numerous ways to represent distributions on the hypersphere. Most of them involve taking a standard distribution from the Euclidian space $\mathcal{R}^n$ and then constraining or projecting them on to the hypersphere $\mathcal{S}^n$. Some of the popular distributions are:

- *Projected Gaussian(s)* on the sphere where standard Gaussian(s) on the tangent space of the hypersphere are projected via central projection, as done in Feiten et al. (2013);
- the *von Mises-Fisher* (vMF) distribution where an isotropic Gaussian on $R^n$ is restricted to the unit hypersphere von Mises (1918);
- The recently developed *Power Spherical* distribution Cao & Aziz (2020), which addresses some of the challenges of the vMF distribution, such as numerical stability and scalability.
- The antipodally symmetric *Bingham* distribution. The antipodal symmetry makes it a suitable distribution to represent quaternions, since quaternions double cover the space of rotations on SO(3) Gilitschenski et al. (2020); Peretroukhin et al. (2020); Srivatsan et al. (2018b;a). However, the Bingham distribution is notorious for its normalization constant that is very hard to compute.

Unfortunately, these distributions are not closed under convolution (i.e. composition of their random variables), thus writing down an closed form diffusion process akin to the Euclidean Gaussian case is intractable. One way to circumvent this problem is to use class of functions that are closed under convolutions on the manifold. An obvious choice is the heat kernel which is the canonical solution to the diffusion equation and is closed under convolutions by definition (Grigoryan, 2009).

