# OpenReview forum: "DIFFUSION GENERATIVE MODELS ON SO(3)"
_ICLR.cc/2023/Conference — Submitted to ICLR 2023_

### Official Review · Reviewer_aEa8 · 2022-10-20

**Confidence:** 5
**Correctness:** 4
**Technical Novelty And Significance:** 3
**Empirical Novelty And Significance:** 3
**Recommendation:** 8

**Clarity, Quality, Novelty And Reproducibility:**

The paper is technically very clear. The materials are presented in a smooth order, making it easy to follow. The mathematical derivations are correct to me.

Results are reproducible. The authors have provided code and sufficient resources in the supplementary materials, which is very welcome.

**Strength And Weaknesses:**

## Strength

The paper is very well-written. It has an excellent review of existing works.

The idea of using Isotropic Gaussian distributions for diffusion generative models on SO(3) has been recently introduced in (Leach et al 2022). However, their approach is restricted to DDPM, while this paper is a more complete treatment of the same topic.

## Weaknesses

While SGM on SO(3) enjoys many properties similar to Euclidean SGM, I think the treatment of DDPM on SO(3) can be further improved. In the paper, the authors mentioned that they did not try out any other distribution families like matrix Fisher distributions or Bingham distributions for the reverse diffusion process but instead used Isotropic Gaussians for convenience. This part can be elaborated further. What convenience do you get by staying with Isotropic Gaussians? Why not try other families? In my view, once you have set up a reverse Markov process as machine-learnable, you are free to choose the distributions that work best for your data. As mentioned in the paper, by staying with Isotropic Gaussians you have to accept an inconvenience in that there is no analytical form for the KL divergence between two distributions of such kind, leading to an extra forward Markov process during training. Why not use a distribution family where the KL divergence has an analytical form?

I think it makes sense to report some comparisons in training times and inference times between SGM on SO(3) and its closest rival RSGM, since RSGM is very close to SGM on SO(3) in spirit but SGM on SO(3) enjoys a much more efficient implementation thanks to the Isotropic Gaussians. Speed-related results are somewhat lacking in the paper.

Equation (14) is somewhat puzzled to me. In the current draft, it looks like the network predicts the mean rotation using the axis-angle representation. However, the axis-angle representation does not work very well when the mean rotation angle approaches $\pi$. Perhaps the authors meant that the function $\mu$ represents the relative rotation (in axis-angle form) from $\mathcal{x}_i$ to the predicted mean rotation, thereby actually having $\mathcal{x}_i$ next to $\exp(\cdot)$ in the equation?


**Summary Of The Paper:**

This paper builds on top of recent, good papers. Diffusion-based generative models have been recently lifted from Euclidean spaces to compact, connected Riemannian spaces (De Bortoli et al, 2022), where given a family of "normal" distributions, forward diffusion processes are proved to admit backward diffusion processes similarly to Euclidean spaces. Normal distributions on those spaces are required to be closed under convolution so that the backward processes can be tractable However, such distributions are typically very cumbersome to deal with as they often exist as infinite serieses. Many approximations have to be introduced. The paper specialises to SO(3), and uses a classical, but less well-known family of "normal" distributions on SO(3) called Isotropic Gaussian distributions $\mathcal{IG}_{SO(3)}$ (Nikolayev & Savyolov, 1970), which is not only closed under convolution but also much easier to work with from the calculus point of view.

The paper goes on and derives an equivalent of score-based generative models (SGMs) on SO(3), and an equivalent of (the spirit of) denoising diffusion probabilistic models (DDPMs) on SO(3). It is shown theoretically that SGM on SO(3) enjoys a lot of similar properties as the Euclidean SGM, as the cumbersome normalizing factor of $\mathcal{IG}_{SO(3)}$ distributions can be mostly avoided.

In contrast, in the DDPM on SO(3) case, because there is no closed form for the distance between two distributions in $\mathcal{IG}_{SO(3)}$, the ELBO cannot be reduced to an analytical form during training.

Experiments on synthetic data comparing the proposed variants showing results favouring SGM on SO(3). However, there is no results on real-world data.

**Summary Of The Review:**

I thank the authors for having submitted this paper. I have had a good read.

The paper has the right motivation. Its theoretical side is very good. There are minor points of improvements for DDPM on SO(3). However, experiments are restricted to synthetic data only, and there is no result related to training time or inference time compared to the existing best method RSGM, which is a bit unfortunate because the paper promotes efficient computations on SO(3).

Nonetheless, I find the pros outweigh the cons.

---

> ### Author Response · Authors · 2022-11-16
> **1st response to referee**
>
> Dear Reviewer,
>
> Thank you for your valuable feedback, which we are working on incorporating into an updated version of our submission.
>
> We are in particular in the process of adding quantitative comparisons of computational costs with respect to RSGM, in terms of the Number of Function Evaluation (NFE) required for sampling (following De Bortoli et al. 2022) but also in terms of training time. As the other reviewers have also pointed out, more numerical experiments on applications would be beneficial and so we are working on adding an example of pose estimation from images on the Symmetric Solids dataset of Murphy et al. 2021 (https://arxiv.org/abs/2106.05965).
>
> Regarding your other comments, indeed you are absolutely right that one is not restricted to the Isotropic Gaussian when parametrizing the DDPM. Any distribution on SO(3) can be used, even a mixture of distributions. Similarly to the Euclidean case, we expect that an isotropic IGSO(3) kernel would be sufficient for many very small denoising steps (i.e. in the limit of a reverse SDE), and that as the number of denoising steps decreases (i.e. coarser updates) more flexibility in the denoising kernels would be beneficial. We are unsure however whether we could obtain an analytic KL given that the forward noise process is still constrained to be an IGSO(3). Time permitting, we will try to include a comparison with alternative kernels.
>
> Finally to your point on Eq. 14, you are absolutely right, our parameterization for \mu_theta is actually in terms of “residual rotations”, which means that the output of the network in practice remains small, away from the \pi discontinuity. We are in the process of improving our technical appendix on network architectures to make these implementation choices clearer. But we are also simultaneously investigating the impact of replacing the axis-angle parametrization for the rotations in the DDPM with continuous parametrizations as described in Zhou et al. 2019 (https://arxiv.org/abs/1812.07035). We will update our main results to the best parameterization and include a comparison in the appendix.
>
> We hope you will find this plan to be able to address your main comments, but please let us know if you think we should consider other experiments as well.

---

> > ### Comment · Reviewer_aEa8 · 2022-11-21
> > **Thank you**
> >
> > Thank you for your response. I look forward to seeing the revised manuscript.
> >
> > Regarding an analytic KL, I think it is worth mentioning in the paper any difficulty you mentioned in the response. To me, the problem is somewhat analogous to dealing with conjugate priors. It can lead to future research.

---

### Official Review · Reviewer_XDGv · 2022-10-25

**Confidence:** 2
**Correctness:** 2
**Technical Novelty And Significance:** 3
**Empirical Novelty And Significance:** Not applicable
**Recommendation:** 5

**Clarity, Quality, Novelty And Reproducibility:**

The paper seems to have novelty in extending diffusion models to SO(3). Previous work (Leach et al. 2022) also uses isotropic Gaussian on SO(3) for the same purpose. But this paper also modifies the loss function, and achieves better results.

The paper is heavy in theory, and therefore, it’s difficult to read for the audience without background. Also because of that, it’s not very clear how to reproduce the implementation.


**Strength And Weaknesses:**

Strength:
- Reformulate diffusion models on SO(3) and the loss functions to train the models.
- The proposed method is better than others in the experiment of synthetic densities.

Weakness
- Experiments are weak, with only three synthetic densities. The paper claims the proposed method has efficient training, but it’s not evaluated in the experiments.
- Applications are also weak. The paper mentioned the proposed method can be used for pose estimation, and claims it’s better than previous work. However, it doesn’t explain how the diffusion generative models can be applied to pose estimation, and there’s no experiment to support the claim.


**Summary Of The Paper:**

The paper extends diffusion generative models to SO(3) manifold. It provides implementations for score-based generative models and denoising diffusion probabilistic models. The proposed methods are applied to synthetic densities on SO(3).

**Summary Of The Review:**

The paper is heavy in mathematical formulation and theory, but is weak in experiments and applications.

---

> ### Author Response · Authors · 2022-11-16
> **1st response to referee**
>
> Dear reviewer,
>
> Thank you for your feedback, we are working on an updated version of our submission that should address your main comments.
>
> In particular, we agree that, as other reviewers have also highlighted, including experiments on pose estimation would be very beneficial to the paper, and as such we are currently working on generating examples of pose estimation from images using the Symmetric Solids dataset from Murphy et al. 2021 (https://arxiv.org/abs/2106.05965) and quantifying the efficiency of our model. We did not include this application initially in our submission as we were concentrating on building an (unconditional) density estimator on the SO(3) manifold, in a similar fashion to works such as Rezende et al 2020 (https://arxiv.org/abs/2002.02428). But once a good neural density estimator exists on a desired manifold, it can be used to build a conditional density estimator. For pose estimation from images, this involves concatenating visual descriptors, computed for instance with a ResNet, to the inputs of our networks.
> We are also in the process of adding quantitative results to compare the computational cost of the methods included in the paper. In particular, we are comparing the Number of Function Evaluation (NFE) required for sampling, following De Bortoli et al. (2022).
>
> Finally, regarding the theory-heavy aspect of our submission, we hope the additional experiments will help balance out the paper with concrete examples of applications, which we agree were lacking.
>
>
> We hope these updates will address your concerns, but please let us know if you think we should consider other experiments as well.

---

### Official Review · Reviewer_aBeQ · 2022-10-28

**Confidence:** 4
**Correctness:** 4
**Technical Novelty And Significance:** 4
**Empirical Novelty And Significance:** 2
**Recommendation:** 5

**Clarity, Quality, Novelty And Reproducibility:**

The paper is well written.

The heat equation on the sphere has been used before for constructing kernels (Multiscale image processing on the sphere, Bulow among others ).

The authors might want to check the "Score-based models detect manifolds" paper by Pidstrigach.


**Strength And Weaknesses:**

+: The main strength of the paper is the use of a heat kernel on SO(3).

+: The authors carefully chose a tractable heat kernel and how to parametrize the isotropic Gaussian as well as explaining its advantages over Bingham and Fisher matrix distribution (being closed wrt convolutions).

+: Thanks to the heat kernel there is no need to solve for the stochastic DE.

+: Isotropic Gaussian enables a closed form in the KL-divergence term of the ELBO.

Authors could respond to the following weaknesses:

-: The mapping from the axis-angle representation to SO(3) is not continuous, and it might affect the stability of the network components of the DE solutions. An example of a continuous mapping is the two first columns of the rotation matrix (a Stieffel manifold) as explained in Zhou et al.'s paper (On the continuity of rotation representations).

-: The experimental evaluation is very limited regarding datasets and evaluation metrics. The significance of density estimators is in capturing multimodal distributions arising in real data. There is abundant real data in 6dof object pose in images and point clouds as well as in all joint orientations in human poses. Many insights can be gained by conditioning the diffusion on such inputs. State-of-the-art approaches like citations Murphy et al. or Mohlin et al. have used such data.



**Summary Of The Paper:**

This paper proposes a generative diffusion model for SO(3) densities using the heat kernel on SO(3). The authors design both a score-based forward pipeline and well as a denoising pipeline for SO(3). Experiments are done on synthetic datasets of rotation samples.


**Summary Of The Review:**

This paper would be an accept if it included experiments conditioned on input images or point clouds. It is the first diffusion on SO(3) as far as I know. Adopting the heat kernel is a great idea, and the implementation of the score-based model and the denoising model are carefully done.

---

> ### Author Response · Authors · 2022-11-16
> **1st response to referee**
>
> Dear reviewer,
>
> Thank you for your feedback. We are working on an updated version of our submission which will address in particular the following points.
>
> Indeed we didn’t initially include experiments on pose estimations from images as our main goal was to develop and demonstrate (unconditional) density estimation on SO(3), , in a similar spirit to Normalizing Flows on Tori and Spheres by Rezende et al 2020 (https://arxiv.org/abs/2002.02428).
> We agree however that additional experiments on conditional density estimation are beneficial and will help to illustrate the interest of this method for specific tasks like pose estimation. We are therefore working on including experiments on the Symmetric Solids (SYMSOL) Dataset of Murphy et al. (2021), using a pre-trained ResNet to produce visual descriptors, which we then use to condition our score network. We are getting very nice preliminary results, and will include those in our revised submission.
>
>
> Regarding the stability of our model given the axis-angle representation, thank you for raising this question, we are investigating the performance of alternative representations (following Zhou et al.) and will update our main results with the best performing one, as well as include a comparison in appendix.
>
> Finally, we are also adding additional citations as suggested, thank you for pointing us to these relevant works.
>
> We hope these updates will address your concerns, but please let us know if you think we should consider other experiments as well.

---

### Author Response · Authors · 2022-11-23
**Response Accompanying Updated Draft**

Dear reviewers,

Thank you again for your feedback. In our updated draft we have made the following changes which we hope will improve the quality of the paper:

1. In Equation (14) of the DDPM we have made it explicit that we parametrize the mean of the reverse transition kernel as a residual with respect to the input rotation.


2. We have experimented with continuous representation for rotations, and indeed found it beneficial to use the 3x3 rotation matrices themselves as inputs to the neural networks, and in the case of the DDPM, to use the 6D representation from Zhou et al. at the output of the network. We have adopted these parameterisations as our baseline, and included comments on these choices in Appendix D
The improvement is small but noticeable in the case of the DDPM compared to our previous results. In the case of the SGM, we did not measure a significant difference in our results.


3. We have included in the main paper a brief illustration of conditional density estimation on SO(3) with our best performing model (SGM) in a pose estimation problem presented in (Murphy et al. 2021). Figure 4 now illustrates the result of conditioning our model on a visual description obtained from images by a ResNet model (as was done in Murphy et al. 2021). More illustrations for different input images are provided in Appendix B.


4. In order to accommodate the additional figure, we have chosen to limit our numerical experiments in the case of the DDPM to the best performing variant (i.e. the Variance Exploding approach akin to Ho et al. 2020) instead of presenting results for both Variance Preserving and Variance Exploding SDEs in our original submission. This allowed us to free the space necessary to add the pose estimation experiment.

5. We have done additional comparisons on the numerical efficiency of our approach compared to the Riemannian Score Generative Model of De Bortoli, et al. (2022). The sampling cost/complexity is equivalent in our case, but training the RSGM model out of the box is 8x slower than our approach in our timing experiments on the same hardware. This is because the RSGM needs to run an SDE forward to produce training examples for their score matching loss. We have added a mention of this timing comparison in the results section.

---

> ### Comment · Reviewer_aEa8 · 2022-11-26
> **Thank you.**
>
> Thank you for the updated draft. Much appreciated.
>
> To my fellow reviewers, what do you think about the authors' responses and the updated draft? It is a good paper, isn't it? If you still have concerns please reply so that the authors can address before time runs out.

---

### Public Comment · ~Haoran_Liu4 · 2023-02-06
**Different definition of epsilon in Eq.4 and Eq.5**

Dear authors,

Thank you for your effort in applying the diffusion model on the SO(3) manifold, and the way to approximate the heat kernel's probability is useful. However, I find that the definition of $\epsilon$ might be different in Eq.4 and Eq.5, as the $\epsilon^2$ in Eq.4 corresponds to $\epsilon$ in Eq.5 as shown in Eq.45 and Eq.47 in [1], so substituting $\epsilon^2$ in Eq.4 with $\epsilon$ might be helpful.

[1] Siegfried Matthies, J. Muller, and G. W. Vinel. On the normal distribution in the orientation space. Textures and Microstructures, 10:77–96, 1988.

---

### Decision · Program_Chairs · 2023-01-20

**Decision:**

Reject

**Justification For Why Not Higher Score:**

The paper does not present enough results to demonstrate the proposed method is useful for challenging real-world cases.

**Justification For Why Not Lower Score:**

N/A

**Metareview: Summary, Strengths And Weaknesses:**

The paper proposes a diffusion model for data live in SO(3). It shows such a diffusion model has a tractable solution when using a heat kernel in SO(3) and supports efficient inference. The paper receives three reviews. One reviewer considers the paper above the bar and voices its acceptance. The reviewer likes the theoretical framework and the presentation. Two reviewers are not convinced that the paper is good enough to make the cut. Their main concerns lie in lacking convincing experiment results to show the proposed framework is actually useful. The authors include experiment results on some limited real-world examples during the rebuttal. However, they are not sufficient to change the opinions of the negative reviewers despite heavy discussion exchanges between the reviewers.

As there is no consensus, the AC must decide. After reading the paper, reviews, and rebuttal, the AC finds the negative reviewers' arguments more convincing. While the paper proposes a nice theoretical framework, not enough effort was dedicated to showing how this formulation works for challenging real-world problems. As a result, the AC would not recommend acceptance of the paper.



**Summary Of Ac-Reviewer Meeting:**

N/A